# Quantifying the impact of solar zenith angle, cloud optical thickness, and surface albedo on the solar radiative effect of Arctic low-level clouds over open ocean and sea ice

Sebastian Becker, André Ehrlich, Michael Schäfer, and Manfred Wendisch

Leipzig Institute for Meteorology (LIM), Leipzig University, Leipzig, Germany

**Correspondence:** Sebastian Becker (sebastian.becker@uni-leipzig.de)

**Abstract.** Arctic low-level clouds play an important although uncertain role in the Arctic climate system. Consequently, their effect on the radiative energy budget (REB) is subject to considerable uncertainty as well. To reduce this uncertainty and to assess the importance of processes driving the cloud radiative effect (CRE), it is crucial to quantitatively disentangle the impact of essential parameters that non-linearly affect the CRE. Therefore, this study uses a CRE parameterization and low-level airborne REB observations in combination with an approach similarly applied in climate dynamics to quantify the contributions of concurrently observed solar zenith angle (SZA), cloud optical thickness, and surface albedo on the solar CRE at the surface. Based on a case study characterized by inhomogeneous cloud and surface conditions in the marginal sea ice zone, it is shown that the surface albedo contributed more than 95 % to the solar CRE difference between open ocean and sea ice. Using the same approach, the analysis is extended to observations from a series of aircraft campaigns and indicates that the variability of the non-cloud properties SZA and surface albedo between seasons and surface types, respectively, has a larger impact on the resulting difference of the solar CRE than the variability of cloud properties.

#### 1 Introduction

The increase of the near-surface air temperature in the Arctic is proceeding at least twice as fast compared to global average values, which is one of the most important signatures of the currently ongoing drastic changes of the Arctic climate system (Wendisch et al., 2023). This rapid transformation results from multiple Arctic-specific processes and feedback mechanisms (e. g., Goosse et al., 2018), amplifying the initial global warming (Serreze and Barry, 2011; Wendisch et al., 2023). Beside Arctic warming, a second obvious indicator of this so-called Arctic amplification is the pronounced decline of the Arctic sea ice, which is particularly prominent during the annual minimum of sea ice extent in September. With respect to the average sea ice extent of the period 1991–2020, Meier et al. (2022) reported a decline of 14% per decade in September, while only 3% per decade were observed in March. The corresponding expansion of open ocean areas has direct and indirect consequences for the radiative energy budget (REB) at the surface. On the one hand, the darker open ocean directly affects the solar REB by increasing the absorption of solar radiation, which leads to an intensified surface warming (surface albedo feedback; e. g., Hall, 2004). On the other hand, stronger upward moisture fluxes over open ocean enhance the formation of clouds (Vavrus et al.,

2011), which belong to the most important modulators of the REB. While this second effect does not play a significant role in summer, cloudiness is expected to increase in autumn (Morrison et al., 2019), concurrent with the strongest sea ice retreat.

Low-level, liquid-containing clouds play a complex role in the Arctic climate system (Wendisch et al., 2019). Their impact on the REB at the surface or the top of the atmosphere (TOA) is defined by the cloud radiative effect (CRE; also referred to as cloud radiative forcing) following Ramanathan et al. (1989):

$$CRE = F_{\text{net,cld}} - F_{\text{net,cf}} \tag{1}$$

$$= (F_{\text{cld}}^{\downarrow} - F_{\text{cld}}^{\uparrow}) - (F_{\text{cf}}^{\downarrow} - F_{\text{cf}}^{\uparrow}). \tag{2}$$

30

45

The net irradiances  $F_{\rm net,cld}$  and  $F_{\rm net,cf}$  represent the REB in cloudy and cloud-free conditions, respectively, and are defined as the difference of the respective downward ( $F_{\rm cld}^{\downarrow}$ ,  $F_{\rm cf}^{\downarrow}$ ) and upward ( $F_{\rm cld}^{\uparrow}$ ,  $F_{\rm cf}^{\uparrow}$ ) irradiances. Here, cloudy conditions refer to situations where a cloud of any fraction is present. While the CRE in the solar spectral range is mostly negative (cooling effect), clouds exert a warming effect in the thermal-infrared (TIR) spectral range of atmospheric radiation. The sign of the total (solar plus TIR) CRE depends on the balance of both components.

A widespread application of the CRE concerns the constraint of the cloud feedback, which is often approximated by the CRE change between two climate states (e. g., Cess et al., 1990; Cesana et al., 2019; Lutsko et al., 2021). However, Soden et al. (2004) demonstrated that this approach does not yield an accurate cloud feedback estimate, because non-cloud properties, such as surface albedo, aerosol particles, or water vapour, can non-linearly affect the CRE even if the cloud characteristics are unchanged. In fact, negative CRE changes, i. e., decreasing CRE, often coincide with positive cloud feedback. These interactions and the opposing effects of solar and TIR CRE complicate accurate estimates of the cloud feedback, which, thus, represents a key source of uncertainty in climate projections (Kay et al., 2016; Choi et al., 2020; Forster et al., 2021). Therefore, it is crucial to precisely investigate the CRE and separate the impacts of different cloud and non-cloud properties on the CRE.

In previous research, the CRE at the surface has been characterized as a complex function of cloud properties, such as cloud optical thickness or height, as well as the concurrent solar zenith angle (SZA) and surface and thermodynamic conditions (e. g., Shupe and Intrieri, 2004). In addition, several studies have investigated the seasonal cycle of the surface CRE using ground-based observations over mostly snow- and ice-covered surfaces across the Arctic (e. g., Intrieri et al., 2002; Miller et al., 2015; Ebell et al., 2020). In contrast to lower latitudes, all Arctic studies identified a total warming effect of clouds on annual average, because the solar cooling effect is limited by the low Sun and the bright surfaces present in the Arctic. Only during summer, a total cooling effect was observed in most cases, when the magnitude of the solar cooling effect surpassed the TIR CRE due to decreasing SZA and surface albedo. Because of this surface albedo dependence of the solar CRE, low-level airborne observations performed during three, seasonally distinct campaigns and analyzed by Becker et al. (2023) revealed a strong total cooling effect of clouds over open ocean as opposed to the adjacent sea ice surfaces. In contrast to the strongly variable solar CRE, the TIR CRE is less affected by seasonal variability, which results from a frequent compensation of increased emission by clouds for warmer temperatures and stronger water vapour absorption below the cloud (Cox et al., 2015; Becker et al., 2023).

Among the rather qualitative studies assessing the CRE, quantitative analyses focussing on the impact of important drivers on the CRE variability are largely lacking. Solely Shupe and Intrieri (2004) estimated sensitivities of the surface CRE with respect to cloud, surface, and thermodynamic properties by applying a simple CRE parameterization and measurements obtained during the shipborne Surface Heat Budget of the Arctic Ocean drift expedition (SHEBA, Uttal et al., 2002). However, these sensitivity estimates do not account for interactions between the considered properties. Yet, a full separation of the relative contributions of different drivers has only been performed for the surface REB. Di Biagio et al. (2012) used a combination of measurements and results from radiative transfer simulations to disentangle the contributions of water vapour, aerosol particles, and multiple scattering from the downward solar irradiance measured at Pituffik Space Base (formerly Thule Air Base), Greenland during cloud-free conditions. They concluded that water vapour usually dominates the solar downward irradiance in cloud-free conditions. Jäkel et al. (2025) analyzed the relative impact of SZA, cloud total water path, and surface albedo on the variability of the solar surface REB over the course of the summer season using Arctic-wide simulations. Applying two regression methods, they identified the SZA to exert the largest impact outside the Central Arctic, while in the Central Arctic the dominant contributor depended on the surface albedo and the corresponding melting stage.

In the present study, a similar parameterization is used in combination with airborne irradiance measurements to quantitatively partition concrete changes of solar CRE into its contributions of SZA, cloud optical thickness, and surface albedo, including the partitioning of interaction terms. Since the TIR CRE is not considered in this study, the term CRE refers explicitly to the solar surface CRE in the following. The measurements and the applied CRE parameterization are described in Sect. 2. The investigations are demonstrated for a case with inhomogeneous cloud and surface conditions, which both affect the evolution of the CRE. Section 3 introduces the case study and disentangles the relative impacts of the drivers on the CRE evolution along the continuous transition across the sea ice edge during this case. Based on a method similar to a technique used in climate dynamics, Sect. 4 furthermore quantifies the relative contributions to a CRE difference between two distinct states, such as different seasons or locations. To demonstrate this approach, the relative contributions of surface albedo and cloud optical thickness are calculated for the CRE contrast between open ocean and sea ice of the example case in Sect. 4.1. In Sect. 4.2, this quantitative method is then applied to cases of CRE contrast between different campaigns and surface types to extend the qualitative analysis by Becker et al. (2023). Section 5 provides the conclusions from this work.

## 2 Observations and solar CRE parameterization

70

#### 2.1 Solar radiation measurements and simulations

The measurements used for this study were obtained in the vicinity of Svalbard during two airborne campaigns, covering different seasons. The Arctic CLoud Observations Using airborne measurements during polar Day campaign (ACLOUD) took place in May/June 2017 (Wendisch et al., 2019; Ehrlich et al., 2019), while the Airborne measurements of radiative and turbulent FLUXes of energy and momentum in the Arctic boundary layer campaign (AFLUX) was performed in March/April 2019 (Mech et al., 2022). Both campaigns employed the research aircraft Polar 5 from Alfred Wegener Institute, Helmholtz Centre for Polar and Marine Research (AWI), which was equipped with an instrumental payload to acquire turbulence, radiation, and

cloud remote sensing data. Cloud microphysical properties were derived from cloud in-situ probes onboard Polar 5 during AFLUX and the additionally deployed AWI aircraft Polar 6 during ACLOUD.

During low-level flight sections, the CRE was retrieved at flight altitude from a combination of solar radiation measurements and radiative transfer simulations. The resulting CRE values are considered representative for the surface since the flight altitude was consistently lower than 250 m and, according to radiative transfer simulations, the corresponding atmospheric impact low-biased the CRE by less than  $2 \, \mathrm{W} \, \mathrm{m}^{-2}$  with respect to the surface. The irradiances  $F_{\mathrm{cld}}^{\downarrow}$  and  $F_{\mathrm{cld}}^{\uparrow}$  in cloudy conditions were measured by broadband Kipp&Zonen CMP-22 pyranometers attached to the aircraft and corrected for aircraft attitude and instrument inertia (Ehrlich et al., 2019). From these observations, the broadband surface albedo  $\alpha$  and  $F_{\mathrm{net,cld}}$  were derived. To obtain  $F_{\mathrm{cf}}^{\downarrow}$ , radiative transfer simulations were performed with the radiative transfer code *uvspec*, which is incorporated in the library for radiative transfer (*libradtran*, Emde et al., 2016). The simulations were initialized with the local SZA calculated from time and position of the aircraft, the measured values of  $\alpha$ , and the observed vertical profiles of air temperature and relative humidity. These profiles resulted from thermodynamic measurements during local aircraft ascents and descents adjacent to the low-level flight sections, which were complemented by the radiosonde observations at Ny-Ålesund (max. 430 km away) for atmospheric layers above the maximum flight altitude. For cloud-free conditions, a comparison between measured and simulated  $F^{\downarrow}$  yields a coefficient of determination  $R^2$  of 0.9971, indicating the accuracy of the simulations.

Further data relevant for the analysis include the sea ice concentration (SIC) and the cloud optical thickness  $\tau$  along the flight track. The SIC was derived from imagery of a downward-looking digital camera equipped with a 180° fish-eye lens based on a sea ice mask (Perovich et al., 2002). To obtain  $\tau$ , a look-up table of simulated  $F_{\rm cld}^{\downarrow}$  created as a function of SZA,  $\alpha$ , and  $\tau$  was applied to the local observations of SZA,  $\alpha$ , and  $F_{\rm cld}^{\downarrow}$  (Stapf et al., 2020). All relevant data are published in Stapf et al. (2021).

## 2.2 Parameterization of solar CRE

The quantitative investigation of the impact of different drivers on the CRE requires a relationship between CRE and cloud and surface properties. Therefore,  $\alpha$  in cloudy conditions as well as the broadband transmissivities of cloud  $\mathcal{T}_{cld}$  and atmosphere (excluding clouds)  $\mathcal{T}_{atm}$  are introduced and obtained from the airborne solar irradiance measurements and the simulations described in Sect. 2.1:

$$\alpha = \frac{F_{\text{cld}}^{\uparrow}}{F_{\text{cld}}^{\downarrow}},\tag{3}$$

$$\mathcal{T}_{\text{cld}} = \frac{F_{\text{cld}}^{\downarrow}}{F_{\text{cf}}^{\downarrow}},$$
 (4)

$$\mathcal{T}_{\text{atm}} = \frac{F_{\text{cf}}^{\downarrow}}{F_0 \cdot \mu}.\tag{5}$$

The denominator in Eq. 5 describes the downward irradiance at the TOA, where  $F_0 = 1361 \, \mathrm{W \, m^{-2}}$  represents the solar constant and  $\mu$  is the cosine of the SZA.

The cloud transmissivity  $\mathcal{T}_{cld}$  accounts for both the direct and diffuse component of the solar downward irradiance. As  $\mathcal{T}_{cld}$  depends on  $\alpha$  and  $\mu$  in addition to the independent cloud property  $\tau$ , it is not an ideally suitable quantity to describe the impact

**Table 1.** Coefficients used in Eqs. 6–8 for the parameterization of  $\mathcal{T}_{cld}$  (Fitzpatrick et al., 2004).

| $a_1$ | $b_1$ | $b_2$   | $b_3$   | $k_1$  | $k_2$  | $k_3$  | c      | d      |
|-------|-------|---------|---------|--------|--------|--------|--------|--------|
| 0.58  | 0.74  | -0.1612 | -0.8343 | 1.9785 | 0.2828 | 2.3042 | 0.1365 | 0.1291 |

of clouds on the CRE. The intensified surface albedo-induced multiple reflections over brighter surfaces cause  $\mathcal{T}_{\rm cld}$  of the same cloud to be higher over sea ice than over open ocean. Therefore,  $\mathcal{T}_{\rm cld}(\mu,\tau,\alpha)$  is expressed by the model-based parameterization of Fitzpatrick et al. (2004):

$$\mathcal{T}_{\text{cld}}(\mu, \tau, \alpha) = \frac{a(\tau) + b(\tau) \cdot \mu}{1 + (c - d \cdot \alpha) \cdot \tau}.$$
(6)

25 The functions  $a(\tau)$  and  $b(\tau)$  in Eq. 6 are given by:

130

135

$$a(\tau) = a_1 + (1 - a_1) \cdot \exp(-k_1 \cdot \tau),$$
 (7)

$$b(\tau) = b_1 \cdot [1 + b_2 \cdot \exp(-k_2 \cdot \tau) + b_3 \cdot \exp(-k_3 \cdot \tau)], \tag{8}$$

while  $a_1$ ,  $b_i$ , c, d, and  $k_i$  (i=1,2,3) in Eqs. 6–8 are coefficients listed in Table 1. These specific values are valid for an effective cloud droplet radius of 8.6  $\mu$ m (Fitzpatrick et al., 2004). Droplets in the order of this size are typical for Arctic low-level clouds (Mioche et al., 2017) and deviations from in droplet size may bias the parameterized  $\mathcal{T}_{\rm cld}$  by less than 2% (Fitzpatrick et al., 2004). Similar to  $\mathcal{T}_{\rm cld}$ , the surface albedo  $\alpha$  is affected by cloud–surface interactions and changes as a function of  $\mu$  and  $\tau$  (e. g., Stapf et al., 2020). However, parameterizations of cloud transmissivity (Fitzpatrick et al., 2004) and surface albedo (Gardner and Sharp, 2010; Jin et al., 2011) suggest that the surface albedo difference between a cloud-free and an opaque-cloud case is weaker than the difference in cloud transmissivity between sea ice and open ocean. Therefore, the change in  $\alpha$  between cloudy and cloud-free conditions is neglected in this study.

Assuming that  $\alpha$  is equal in cloudy and cloud-free conditions, inserting Eqs. 3–5 into Eq. 2 and replacing  $\mathcal{T}_{\rm cld}$  by the parameterization of Eq. 6 yields an expression for the solar CRE that is dependent on  $\mathcal{T}_{\rm atm}$ ,  $\mu$ ,  $\tau$ , and  $\alpha$ :

$$CRE(\mu, \tau, \alpha) = F_0 \cdot \mathcal{T}_{atm} \cdot \mu \cdot \mathcal{T}_{cld}(\mu, \tau, \alpha) - F_0 \cdot \mathcal{T}_{atm} \cdot \mu \cdot \mathcal{T}_{cld}(\mu, \tau, \alpha) \cdot \alpha - F_0 \cdot \mathcal{T}_{atm} \cdot \mu + F_0 \cdot \mathcal{T}_{atm} \cdot \mu \cdot \alpha. \tag{9}$$

This expression is identical to the parameterization used by Shupe and Intrieri (2004, their Eq. 5). For  $\mathcal{T}_{\rm atm}$ , a constant value of 0.75 is assumed, corresponding to the mean value of all observations collected during AFLUX and ACLOUD in cloudy conditions with  $\tau$  larger than 1.125 (this threshold corresponds to a liquid water path of 5 g m<sup>-2</sup> assuming an effective droplet radius of 8 µm). This assumption appears serious, but is justified by the excellent correlation between the observed and the parameterized CRE (Eq. 9) that is demonstrated by Fig. 1 and indicated by the coefficient of determination  $R^2 = 0.9939$ . This overall high accuracy of the CRE parameterization results from the fact that both the CRE and the regressors were largely retrieved from the same irradiance quantities. Minor deviations are primarily caused by the variation of  $\mathcal{T}_{\rm atm}$  between 0.63 and 0.82, which, similar to  $\mathcal{T}_{\rm cld}$ , depends on  $\mu$ ,  $\alpha$ , and the optical thickness of the cloud-free atmosphere that is affected by aerosol particles and trace gases. However, the following quantification of the drivers' impact on the CRE is hardly limited by the applied assumptions.

**Figure 1.** Two-dimensional probability density function depending on parameterized solar CRE (Eq. 9) and observed solar CRE considering all cloudy observations ( $\tau$  larger than 1.125) of AFLUX and ACLOUD. The dashed line marks the 1:1-line.

## 3 Relative contributions for continuous observations

# 150 **3.1** Case study

The approach to disentangle the impact of various drivers on a continuous transition of the CRE, e.g., from open ocean to sea ice, is demonstrated for a case study, which is based on a flight section with variable cloud and surface conditions performed during AFLUX on 4 April 2019. The basic weather situation during this flight is illustrated in Fig. 2. Both at the surface and the 850 hPa pressure level, a low-pressure system was located north-west of the Fram Strait, while high pressure was present towards south-east. At the surface, this constellation caused a southerly advection of warm air west of Svalbard, while south-westerly wind at 850 hPa pushed a cloud with inhomogeneous optical thickness over the sea ice edge. Roughly parallel to the 850 hPa isohypses and across the sea ice edge, a flight leg was set up that was flown four times by Polar 5 in different altitudes. The second leg, headed from north-east to south-west, was performed at low altitude and offered the subset of observations used for the analysis of this example case.

Figure 3a illustrates the evolution of  $\mu$ ,  $\tau$ , and  $\alpha$  (consistent colour coding used throughout the study) along the low-level flight leg as a function of the geographic latitude. To reduce small-scale variability, all time series are smoothed with a two-minute Hann window. The corresponding SIC is shown in Fig. 3b. The SZA ( $\mu$ ) varied only weakly between 74° (0.276) at the south-western and 75.3° (0.254) at the north-eastern end of the flight leg. In contrast, the variability of  $\alpha$  and  $\tau$  was substantial. The surface albedo increased from values less than 0.1 over open ocean to almost 0.9 over sea ice with an enhanced variability in the marginal sea ice zone (MIZ). The cloud optical thickness ranged between 5 and 40 and the optically thickest clouds were

**Figure 2.** True-colour satellite image (composite of MODIS channels 1, 4, and 3) observed on 4 April 2019, 10:15 UTC and overlaid by mean sea level pressure and 850 hPa geopotential from ERA5 reanalysis (Hersbach et al., 2020), and the 15 % isoline of satellite-derived sea ice concentration (Spreen et al., 2008). Additionally, the flight track and the low-level flight leg are highlighted.

observed in the central part of the flight leg. Beside this intermediate cloud thickening,  $\tau$  was higher and ranged up to 22 over open ocean, while it did not exceed 10 over sea ice.

Due to the weak variability of the SZA indicated by Fig. 3a, the CRE change is not significantly driven by the SZA in the present case study. Therefore,  $\mu$  in Eq. 9 is fixed to its mean value along the low-level flight leg ( $\bar{\mu}=0.264$ , corresponding to SZA = 74.7°). Additionally,  $\mathcal{T}_{\rm atm}$  is adjusted to the mean value resulting from this subset of observations only (0.72). The calculated CRE is represented by the black line in Fig. 3a and resembles the observed CRE (red line) with  $R^2=0.9978$ .

## 3.2 Method and application to the case study

For continuous observations with weak differences of the drivers between neighbouring data points, the total differential of Eq. 9 (neglecting the SZA dependence), with

$$dCRE = S_{\tau}(\tau, \alpha) \cdot d\tau + S_{\alpha}(\tau, \alpha) \cdot d\alpha,$$
 (10)

Figure 3. Evolution of (a) observed and parameterized (Eq. 9) solar CRE (left y-axis) as well as  $\mu$ ,  $\tau$  and  $\alpha$  (right y-axes) smoothed with a 2-minute Hann window, and (b) sea ice concentration (SIC) along the low-level flight leg performed on 4 April 2019 as a function of geographic latitude. Time series of (c) temporal solar CRE gradient (black) as well as its absolute contributions of  $\tau$  and  $\alpha$ , and (d) the relative impact of  $\tau$  and  $\alpha$  to the solar CRE change, applying the same colour coding as in (a). The time series in (d) are additionally smoothed with a 30-second Hann window and the background colour indicates the dominant CRE driver. See text for more details.

yields an accurate result for the corresponding change of the CRE. The terms on the right-hand side of Eq. 10 represent the absolute contributions of  $\tau$  and  $\alpha$  to the CRE change, which are determined by both the sensitivities of the CRE with respect to  $\tau$  ( $S_{\tau}$ ) and  $\alpha$  ( $S_{\alpha}$ ) and the absolute change of these parameters ( $d\tau$ ,  $d\alpha$ ). The sensitivity coefficients, given by

$$S_{\tau}(\tau, \alpha) = \frac{\partial \text{CRE}}{\partial \tau} \text{ and}$$
 (11)

$$S_{\alpha}(\tau, \alpha) = \frac{\partial \text{CRE}}{\partial \alpha},$$
 (12)

both depend on  $\tau$  and  $\alpha$  and are discussed in detail in Appendix A. Along the flight leg of the example case, the results of the separated contributions are shown in Fig. 3. The temporal changes of the absolute contributions of  $\tau$  and  $\alpha$  are illustrated

in Fig. 3c and indicate their respective tendency to the CRE transition. The precise calculation of relative contributions of  $\tau$  and  $\alpha$  as the ratio of these tendencies to the CRE change fails when the latter approaches zero. Therefore, the relative impacts shown in Fig. 3d are calculated as the absolute contributions relative to the sum of the magnitudes of all absolute contributions. The positive or negative sign indicates whether a relative impact agrees with or opposes the CRE change, respectively. This measure is assured to not exceed 1 in magnitude. Nevertheless, peaks with short periods of sign conversion still occur in the neighbourhood of zeros of the CRE change and require an additional smoothing with a 30-second Hann window.

The generally decreasing  $\tau$  south of 79.7° N and between 80.0° and 80.1° N caused a positive tendency to the CRE at most points, while the intermediate increase of  $\tau$  tended to mostly decrease the CRE (green line in Fig. 3c). North of 80.1° N, where  $\tau$  was rather constant, its contribution fluctuated between positive and negative tendencies depending on the exact gradient, but revealed only a small value on average. Likewise, the surface albedo contribution (blue line in Fig. 3c) oscillated around zero over open ocean, but showed more persistent periods of positive and negative tendencies to the CRE in the MIZ between 79.8° and 79.9° N and between 79.9° and 80.05° N, respectively. Towards the sea ice, the variable but broadly increasing  $\alpha$  resulted in a fluctuating contribution with a positive tendency on average.

The quantity with the largest relative impact indicates the dominant driver of the CRE evolution, which is highlighted by the background colour in Fig. 3d. The frequent green background for latitudes less than 79.7° demonstrates that the CRE change was mostly controlled by the evolution of  $\tau$  over open ocean, where the rather low surface albedo change caused only small positive or negative impacts. In contrast,  $\alpha$  largely drove the CRE as soon as the MIZ was reached (dominant blue background for higher latitudes). North of 80.1° N, the CRE change in Fig. 3c basically follows the contribution term of  $\alpha$  due to the weak cloud variability. Despite the intensive cloud thickening,  $\tau$  did not significantly affect the CRE in the central part of the leg due to the weak sensitivity of the CRE on  $\tau$  for optically thick clouds. Only towards the end of the following cloud thinning between 80.0° and 80.05° N,  $\tau$  briefly dominated the CRE change.

## 4 Relative contributions between states

205

210

Due to the assumption of infinitesimal differences in Eq. 10, the approach described in Sect. 3.2 may lead to significant uncertainties if the differences of  $\tau$  and  $\alpha$  between two data points become too large. This is particularly the case, when the non-constant sensitivity coefficients  $S_{\tau}(\tau,\alpha)$  and  $S_{\alpha}(\tau,\alpha)$  (see Appendix A) vary significantly between the two points, causing a considerable discrepancy between the CRE change (left-hand side of Eq. 10) and the sum of the absolute contributions (right-hand side of Eq. 10). In this case, another method, which is proposed in the following and applicable to any point-to-point difference, may be considered. This method is likewise suitable for disentangling the contributions of the drivers to a CRE change between two isolated states, such as different points in time or location. Based on the parameterization of Eq. 9, the contribution of a driver is quantified by the partial CRE difference resulting from a sole change of the associated variable between the two considered points, while the other variables are kept constant. In parts, this approach is similar to the approximated partial radiative perturbation (APRP) technique applied in climate dynamics, where a parameterization of the solar REB at the TOA is used to decompose the solar REB difference between two climate states into the contributions

Figure 4. Solar CRE parameterized with Eq. 9 as a function of  $\tau$  and  $\alpha$ . The symbols indicate the median states  $(\alpha_1, \tau_1)$  over open ocean and  $(\alpha_2, \tau_2)$  over sea ice calculated for the case study on 4 April 2019. The blue and green numbers (all in W m<sup>-2</sup>) quantify the partial solar CRE change along the respective lines. The red numbers represent the finally obtained absolute contributions of  $\tau$  and  $\alpha$ , corresponding to the partial solar CRE differences at the evaluation point  $(\alpha_e, \tau_e)$ . The evaluation point is determined by the intersection of two criteria: first, it must lie on the black solid line connecting the two states and, second, it must satisfy Eq. 13, which is the case for all  $(\alpha, \tau)$  along the black dashed line. See text for more details.

of various feedback mechanisms (Taylor et al., 2007). However, due to the different quantities and fields of application, the contributions calculated here are not comparable to the results from Taylor et al. (2007).

## 4.1 Method based on case study

The applied method is demonstrated for the case study introduced in Sect. 3.1. Figure 4 illustrates the CRE calculated from Eq. 9 as a function of  $\tau$  and  $\alpha$ . The two highlighted states are defined by the median values obtained from the observations over open ocean (SIC less than 5%) and sea ice (SIC larger than 95%). These median values of  $\tau$  and  $\alpha$  amount to  $\tau_1 = 8.5$  and  $\alpha_1 = 0.10$  over open ocean (dot in Fig. 4) and  $\tau_2 = 7.0$  and  $\alpha_2 = 0.78$  over sea ice (cross in Fig. 4). With a stronger cooling effect for higher  $\tau$  and lower  $\alpha$ , the calculated CRE over open ocean and sea ice are  $-144.2 \,\mathrm{W \, m^{-2}}$  and  $-11.9 \,\mathrm{W \, m^{-2}}$ , respectively. These values compare well to the observed median CRE values of  $-144.3 \,\mathrm{W \, m^{-2}}$  and  $-13.3 \,\mathrm{W \, m^{-2}}$  and produce a CRE difference of  $\Delta$ CRE = 132.3 W m<sup>-2</sup>.

The decomposition of  $\Delta CRE$  into partial CRE differences that only account for a change in  $\tau$  or  $\alpha$  between the two states is given by:

$$\Delta CRE = \Delta CRE_{\Delta \tau}(\alpha) + \Delta CRE_{\Delta \alpha}(\tau), \tag{13}$$

which is equivalent to integrating Eq. 10 between the states. For example, the partial CRE difference  $\Delta \text{CRE}_{\Delta\tau}(\alpha)$  represents the CRE contrast resulting from a change  $\Delta\tau$  from  $\tau_1$  to  $\tau_2$  at any constant  $\alpha$ . Due to the non-linear sensitivity of the CRE on both  $\tau$  and  $\alpha$ ,  $\Delta \text{CRE}_{\Delta\tau}(\alpha)$  and  $\Delta \text{CRE}_{\Delta\alpha}(\tau)$  depend on the concrete value that  $\alpha$  and  $\tau$ , respectively, are fixed to. These non-linearities are indicated by the pairs of green and blue numbers in Fig. 4. Despite an identical  $\Delta\tau$ , the associated  $\Delta \text{CRE}_{\Delta\tau}$  is 8.3 W m<sup>-2</sup> if  $\alpha$  corresponds to  $\alpha_1$ , but only 0.6 W m<sup>-2</sup> for  $\alpha_2$ . Similarly,  $\Delta \text{CRE}_{\Delta\alpha}$  amounts to 131.6 W m<sup>-2</sup> if  $\tau = \tau_1$  and 124.0 W m<sup>-2</sup> for  $\tau = \tau_2$ . Consequently, for neither  $(\alpha_1, \tau_1)$  nor  $(\alpha_2, \tau_2)$ , the partial CRE differences do exactly add up to  $\Delta \text{CRE}$  in Eq. 13. Therefore, the approach suggested in the following, which was not considered by the APRP method (Taylor et al., 2007), identifies the values  $(\alpha_e$  and  $\tau_e$ ) that precisely satisfy Eq. 13. This pair of values is referred to as evaluation point in the following.

Since Eq. 13 is underconstrained with the two unknown variables  $\alpha$  and  $\tau$ , the possible solutions to it are distributed along the black dashed line in Fig. 4 and include both  $(\alpha_1, \tau_2)$  and  $(\alpha_2, \tau_1)$ . However, the fraction of the partial CRE differences with respect to the total CRE difference (i. e., the relative contributions) are not identical for these solutions. To obtain a unique pair of relative contributions, an additional criterion is introduced, which requires  $(\alpha_e, \tau_e)$  to lie on the straight connection line between the two states (black solid line in Fig. 4), parameterized as

$$\begin{pmatrix} \tau \\ \alpha \end{pmatrix} = \begin{pmatrix} \tau_1 \\ \alpha_1 \end{pmatrix} + s \begin{pmatrix} \tau_2 - \tau_1 \\ \alpha_2 - \alpha_1 \end{pmatrix}.$$
 (14)

By inserting Eq. 14 into Eq. 13, this requirement yields a solution for the parameter s that is used to calculate the final evaluation point  $(\alpha_e, \tau_e) = (0.49, 7.7)$ . For these values, the partial CRE differences eventually quantify the absolute contributions of cloud and surface, which amount to 4.3 W m<sup>-2</sup> and 127.9 W m<sup>-2</sup> (red numbers in Fig. 4) and correspond to relative contributions of 3.3 % and 96.7 %, respectively.

## 4.2 Application to different seasons and surface types

The method described in Sect. 4.1 can be used to quantify the contributions of the various drivers to a CRE difference between two arbitrary states. For four additional cases, this section calculates the contributions of  $\mu$ ,  $\tau$ , and  $\alpha$  to the CRE differences between surface types and seasonally different campaigns, which were qualitatively discussed by Becker et al. (2023). Since the SZA variation between these observations was significant, the dependence of the SZA is included in these calculations. Thus, a contribution term of  $\mu$  is added to Eq. 13, such that

$$\Delta CRE = \Delta CRE_{\Delta\mu}(\tau, \alpha) + \Delta CRE_{\Delta\tau}(\mu, \alpha) + \Delta CRE_{\Delta\alpha}(\mu, \tau), \tag{15}$$

Figure 5. Median values of (a)  $\mu$ , (b)  $\tau$ , and (c)  $\alpha$  and (d) the resulting solar CRE parameterized with Eq. 9 for the two states compared for the respective case. The cases are labelled such that the numbers before and after " $\rightarrow$ " correspond to the state represented by the left and right bar, respectively; see legend for the numbers assigned to each state. The red crosses in (d) denote the observed solar CRE for each state. The coordinates  $\mu_e$ ,  $\tau_e$ , and  $\alpha_e$  of the evaluation point obtained for each case are marked with the red diamonds in (a–c). (e) Absolute (coloured bars) and relative (numbers) contributions to the CRE difference between the states (black bar) of each case.

and the vectorial Eq. 14 now reads:

$$\begin{pmatrix} \mu \\ \tau \\ \alpha \end{pmatrix} = \begin{pmatrix} \mu_1 \\ \tau_1 \\ \alpha_1 \end{pmatrix} + s \begin{pmatrix} \mu_2 - \mu_1 \\ \tau_2 - \tau_1 \\ \alpha_2 - \alpha_1 \end{pmatrix}.$$
 (16)

For all cases and the corresponding states, the median values of  $\mu$ ,  $\tau$ , and  $\alpha$ , the calculated CRE and the retrieved contributions are summarized in Fig. 5. The two leftmost cases investigate the CRE difference between open ocean and sea ice for AFLUX and ACLOUD, while the remaining cases quantify the contributions between the two campaigns, separately for open ocean and sea ice. The significantly different  $\alpha$  (Fig. 5c) dominates the CRE difference between open ocean and sea ice with a relative contribution of at least 84.7 %. Only during AFLUX, the lower SZA over open ocean compared to sea ice (75.5° vs.

79.6°) significantly contributed to the CRE difference with 12.9%. Comparing AFLUX and ACLOUD, the seasonally different SZA (Fig. 5a) contributes most to the CRE difference, but seasonal changes in  $\alpha$  are not negligible. Especially over sea ice, the snow albedo was decreased by melting and contributed 35.9% to the CRE difference, while the dominant SZA contribution of 47.9% was relatively weak. However, note that the neglect of the albedo change between cloudy and cloud-free conditions overestimated the relative contribution of  $\alpha$  to this CRE difference. In contrast, the 17.9% contribution of  $\alpha$  over open ocean is likely an artifact of sea smoke below the aircraft, which increased the measured albedo during AFLUX (Becker et al., 2023). If the sea smoke had not been present and the open ocean albedo had revealed a typical value of 0.06, the relative contribution of  $\alpha$  would have reverted to -2.0% in favour of the SZA contribution. The states of all cases were dominated by optically thick clouds, to which the sensitivity of the CRE is weak (see Appendix A). Therefore, the relative contributions of  $\tau$  are generally low. During ACLOUD,  $\tau$  negatively contributed to the CRE change due to optically thicker clouds over sea ice. The largest difference in  $\tau$  occurred over sea ice between AFLUX and ACLOUD (Fig. 5b), resulting in the highest relative contribution of  $\tau$  between two states with 16.2%.

## 5 Summary and conclusions

290

Future changes of the Arctic climate are expected to alter the properties of clouds, resulting in a modification of the cloud radiative effect (CRE). Since this CRE modification is similarly affected by concurrent changes in non-cloud properties, such as surface albedo, separating the cloud and various non-cloud contributions composing the CRE is crucial to accurately identify, represent, and disentangle different cloud-involving interactions and their relative importances. Based on Shupe and Intrieri (2004), this study developed a simple but accurate parameterization of the solar CRE, which is applied to airborne radiation observations to quantitatively investigate the impact of concurrently observed SZA, surface albedo α, and cloud optical thickness τ on the solar surface CRE. These investigations were largely based on an appropriate case study with inhomogeneous cloud and surface conditions in the vicinity of the marginal sea ice zone. Since the SZA was almost constant around 75°, its impact was negligible for this example case. For continuous observations, the impact of each driver on the CRE evolution was determined by the respective term of the total differential of the CRE. During the example case, the surface transition from open ocean to sea ice clearly dominated the solar CRE despite a significant intermediate cloud thickening, which was largely inefficient due to the weak CRE sensitivity on τ for optically thick clouds.

Since the method using the total differential can lead to significant uncertainties for too large changes of the drivers, an alternative approach to disentangle their contributions was introduced. This decomposition method is similar to the approximated partial radiative perturbation technique (Taylor et al., 2007) and also applicable to partition the CRE difference between two distinct states into the contributions of the drivers. For the example case, this method revealed that the contrasting surface albedo contributed more than 95 % to the solar CRE difference between open ocean and sea ice, while the cloud impact was weak. Using observations from an airborne spring and summer campaign, the method was applied to additionally calculate relative contributions for CRE differences between different surface types and seasons. The quantified contributions confirmed the qualitative assessment of Becker et al. (2023). The solar CRE difference between open ocean and sea ice is at least 84 % due

to the surface albedo contrast, while the SZA difference contributed more than half to the CRE change from spring to summer. The cloud impact itself was found to be low in all cases, corroborating the frequent dominance of non-cloud properties for the CRE. Nevertheless, the conclusions are based on limited airborne samples, which might be biased by flight strategy. It would be useful to apply the described method to further, statistically more robust data sets to extensively investigate the impact of the changing cloud and environmental properties on the solar CRE. The general approach used in this study not limited to the Arctic. Since the method is universally applicable to quantify the contributions of drivers to any given CRE difference, it could also be used to assess how the importance of certain drivers differs, e. g., between the polar regions and the mid-latitudes, where surface albedo contrasts are usually weaker. Furthermore, modelling could possibly benefit from quantifying the contributions of the drivers to a potential CRE bias, which can help to evaluate for which parameters an accurate representation in the model is most crucial. Finally, to disentangle the full impacts of cloud and non-cloud properties on the total CRE, a comparable analysis for the thermal-infrared (TIR) CRE would be required. However, a similarly simple method is challenging due to the strong dependence of the TIR CRE on profiles of temperature, water vapour, and clouds.

## Appendix A: Sensitivity of solar CRE

310

Based on Eqs. 11 and 12 and the fixed  $\bar{\mu}=0.264$ , the sensitivities  $S_{\tau}(\tau,\alpha)$  and  $S_{\alpha}(\tau,\alpha)$  are calculated for a wide range of  $\tau$  and  $\alpha$  and illustrated in Fig. A1. The sensitivity of the CRE with respect to  $\tau$  is negative (Fig. A1a), indicating an enhanced cooling effect with increasing  $\tau$ . The CRE is particularly sensitive to optically thin clouds ( $\tau$  less than 2) over open ocean, where the magnitude of  $S_{\tau}$  generally exceeds  $20\,\mathrm{W\,m^{-2}}$  per unit of  $\tau$ . For the same values of  $\tau$  over sea ice,  $S_{\tau}$  is reduced due to the weaker solar cooling effect compared to open ocean. Optically thick clouds with  $\tau$  larger than 10 cause a weak magnitude of  $S_{\tau}$ , not exceeding  $5\,\mathrm{W\,m^{-2}}$  per unit of  $\tau$  over open ocean and  $1\,\mathrm{W\,m^{-2}}$  per unit of  $\tau$  over sea ice. Generally, the magnitude of  $S_{\tau}$  increases with both decreasing  $\tau$  and decreasing  $\alpha$ .

The sensitivity of the CRE with respect to  $\alpha$  (Fig. A1b) strengthens with increasing  $\tau$  and decreasing  $\alpha$  and the positive values express a larger CRE (weaker cooling effect) for higher  $\alpha$ . In optically thick cloud conditions over open ocean, the CRE is particularly sensitive to surface albedo changes ( $S_{\alpha}$  larger than 200 W m<sup>-2</sup> per unit of  $\alpha$ ). However, clouds with  $\tau$  equal to 1 are sufficient for a minimum  $S_{\alpha}$  of 50 W m<sup>-2</sup> per unit of  $\alpha$ .

Data availability. The data analyzed in this manuscript are publically available on the PANGAEA database (Stapf et al., 2021, https://doi. pangaea.de/10.1594/PANGAEA.932010).

*Author contributions.* SB developed the method, analyzed the data for this study, and drafted the article. MW and AE designed the experimental basis of this study. All authors contributed to discussion and interpretation of the results and the editing of the article.

**Figure A1.** Sensitivity of the solar CRE with respect to (a) cloud optical thickness  $(S_{\tau})$  and (b) surface albedo  $(S_{\alpha})$  as a function of  $\tau$  and  $\alpha$ , calculated for the constant  $\bar{\mu}$  of 0.264 (corresponding to SZA of 74.7°).

Competing interests. The authors declare that they have no conflict of interest.

Acknowledgements. We gratefully acknowledge the funding by the Deutsche Forschungsgemeinschaft (DFG, German Research Foundation)

- Projektnummer 268020496 – TRR 172, within the Transregional Collaborative Research Center "ArctiC Amplification: Climate Relevant Atmospheric and SurfaCe Processes, and Feedback Mechanisms (AC)<sup>3</sup>. This work was funded by the Open Access Publishing Fund of Leipzig University supported by the DFG within the program Open Access Publication Funding.

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
