# Peer review of "Quantifying the impact of solar zenith angle, cloud optical thickness, and surface albedo on the solar radiative effect of Arctic low-level clouds over open ocean and sea ice"

_EGUsphere, 2025_

## Author Comment (AC1)

**Replies to comments of Review #1**

We would like to thank the reviewers for their constructive feedback and appreciate their time for extensively reading and commenting on the submitted manuscript. Our replies to the referees' comments are structured as follows:

*Referee's comments in italic – line numbers according to initially submitted manuscript*
Authors' responses in roman – line numbers according to adjusted manuscript.
**Citations from the initial and the adjusted manuscript are given in bold.**

Beyond edits related to the reviews, we applied additional revisions to some text passages after carefully going through the manuscript. For these changes, we refer to the track changes file.

*"Quantifying the impact of solar zenith angle, cloud optical thickness, and surface albedo on the solar radiative effect of Arctic low-level clouds over open ocean and sea ice" by Becker et al. investigates the relative contributions of cloud properties (summarized as optical thickness) and surface albedo to solar cloud radiative effects using observations made from aircraft during the ACLOUD and AFLUX campaigns. The study concludes that surface albedo overwhelmingly dominates over cloud properties in the difference in observed CRE between ocean and sea ice domains. The study presents some interesting concepts and results. I think it will suitable for ACP with a revision that addresses concerns that are both technical in nature and relate to the overall scoping of the study's motivation and interpretation.*

*General Concerns:*

1. *The introduction needs work. It provides a review of the cloud radiative forcing in the infrared, with discussion of solar effects being absent or incidental. It then says the present study doesn't analyze the infrared, only solar. Seems like the intro should focus on the current state of, and outstanding gaps in knowledge of solar radiation.*

Thanks for identifying this rather one-sided discussion in the introduction. Usually, the studies cited in the literature review of the CRE investigated both the solar and the TIR CRE, such that there are no larger gaps in knowledge of the solar compared to the TIR CRE. Therefore, we tried to balance the discussion by including the solar CRE in the introduction:

**"In previous research, the CRE at the surface has been characterized as a complex function of cloud properties, such as optical thickness or height, as well as the concurrent solar zenith angle (SZA) and surface and thermodynamic conditions (e. g., Shupe and Intrieri, 2004). In addition, several studies have investigated the seasonal cycle of the surface CRE using ground-based observations over mostly snow- and ice-covered surfaces across the Arctic (e. g., Intrieri et al., 2002; Miller et al., 2015; Ebell et al., 2020). In contrast to lower latitudes, all Arctic studies identified a total warming effect of clouds on annual average, because the solar cooling effect is limited due to the low Sun and the bright surfaces present in the Arctic. Only during summer, a total cooling effect was observed in most cases, when the magnitude of the solar cooling effect surpassed the TIR CRE due to decreasing SZA and surface albedo. Because of this surface albedo dependence of the solar CRE, low-level airborne observations performed during three, seasonally distinct campaigns and analyzed by Becker et al. (2023) revealed a strong total cooling effect of**

clouds over open ocean as opposed to the adjacent sea ice surfaces. In contrast to the strongly variable solar CRE, the TIR CRE is less affected by seasonal variability, which results from a frequent compensation of increased emission by clouds for warmer temperatures and stronger water vapour absorption below the cloud (Cox et al., 2015; Becker et al, 2023).

Among the rather qualitative studies assessing the CRE, quantitative analyses focussing on the impact of important drivers on the CRE variability are largely lacking. Solely Shupe and Intrieri (2004) estimated sensitivities of the surface CRE with respect to cloud, surface, and thermodynamic properties by applying a simple CRE parameterization and measurements obtained during the shipborne Surface Heat Budget of the Arctic Ocean drift expedition (SHEBA, Uttal et al., 2002). However, these sensitivity estimates do not account for interactions between the considered properties. Yet, a full separation of the relative contributions of different drivers has only been performed for the surface REB." (lines 44–62)

2. *The study is motivated by the need to constrain cloud feedback estimates, but it does not investigate cloud feedbacks at all, despite what the study sometimes suggests, such as at the top of section 4. Fundamental to the feedback concept is a change in cloud properties as a response to some climate forcing, which may, for example, be measured by cloud forcing. Here the concept is conflated with the difference in cloud forcing between when a cloud is over ocean and when it is over ice. Therefore, the "distinct states" referred to throughout the text are not analogous to Taylor et al., Soden et al. etc. Sometimes, these are referred to as "climate states" (e.g., L200), which is more consistent with the referenced literature, and other times as "locations", "seasons" (L75), or just "states" (throughout), which is less consistent. Regardless, these are not interchangeable concepts. I like the objective here of trying to understand how varying surface and cloud properties result in the CRE that is observed, but the study should not overstate its connection to the problem of cloud feedbacks.*

It is correct that our study does not investigate the cloud feedback and we also did not intend to do so. Our motivation of the cloud feedback constraint was rather meant as an example for application of the CRE. However, the fact that the CRE, due to its dependence on non-cloud properties, is an inaccurate measure for cloud feedback should motivate the need to disentangle the various drivers of the CRE. To make our intention clearer, we rewrote the cloud feedback part of the motivation and put a stronger emphasis on the CRE and its drivers, but still using the cloud feedback example:

"A widespread application of the CRE concerns the constraint of the cloud feedback, which is often approximated by the CRE change between two climate states (e. g., Cess et al., 1990; Cesana et al., 2019; Lutsko et al., 2021). However, Soden et al. (2004) demonstrated that this approach does not yield an accurate cloud feedback estimate, because non-cloud properties, such as surface albedo, aerosol particles, or water vapour, can non-linearly affect the CRE even if the cloud characteristics are unchanged. In fact, negative CRE changes, i. e., decreasing CRE, often coincide with positive cloud feedback. These interactions and the opposing effects of solar and TIR CRE complicate accurate estimates of the cloud feedback, which, thus, represents a key source of uncertainty in climate projections (Kay et al., 2016; Choi et al., 2020; Forster et al., 2021). Therefore, it is

**crucial to precisely investigate the CRE and separate the impacts of different cloud and non-cloud properties on the CRE."** (lines 36–43)

Regarding Section 4: We understand that CRE and cloud feedback are different concepts and we never intended to provide cloud feedback estimates with this study. However, in our opinion, it is still feasible to apply a decomposition method similar to the approximated partial radiative perturbation (APRP) technique (Taylor et al., 2007) to our problem, although the original method was developed to disentangle the shortwave (solar) components of different (not only cloud) feedbacks. The common goal of the APRP and our application is to partition a change in radiation budget between two states into the contributions of different drivers. The radiation budget is given as a quantifiable function of the drivers and the contribution of a driver is quantified by simultaneously only changing the value of this single driver between state 1 and state 2, while keeping the others constant. The analogies between both applications are summarized in the following table:

|  | Original (Taylor et al., 2007) | This study |
|---|---|---|
| radiation budget difference | net shortwave radiative forcing and response at the TOA
→ together simply the difference in TOA net shortwave radiation | difference in solar surface CRE |
| drivers of radiation budget difference | shortwave feedbacks due to changing
- surface albedo
- cloud properties
- atmosphere (e. g., water vapour) | changing
- surface albedo
- cloud optical thickness
- solar zenith angle |
| states | control vs. perturbed climate or
two different times ... | sea ice vs. open ocean or
two different seasons, locations, ... |

The only purpose of introducing the APRP method in the introduction to Section 4 is highlight the similarities and differences between the APRP and our application. To make this intention more obvious, we shortened the description of the APRP method and put more emphasis on our application in the revised version of this paragraph:

**"Based on the parameterization of Eq. 9, the contribution of a driver is quantified by the partial CRE difference resulting from a sole change of the associated variable between the two considered points, while the other variables are kept constant. In parts, this approach is similar to the approximated partial radiative perturbation (APRP) technique applied in climate dynamics, where a parameterization of the solar REB at the TOA is used to decompose the solar REB difference between two climate states into the contributions of various feedback mechanisms (Taylor et al., 2007). However, due to the different quantities and fields of application, the contributions calculated here are not comparable to the results from Taylor et al. (2007)."** (lines 211–217)

3. *How many total hours of observations are included here? How many unique clouds are sampled? I suspect the samples are limited. While there are some nice methods and analyses here, we are still working with a few case studies. Therefore, while I agree that similar analyses in the infrared is a good suggestion (L274), the main recommendation for the community should be more studies focused on how cloud and environmental properties combine to produce cloud forcing without the implicit suggestion this is the final word on the solar component. Be careful not to overstate.*

It is correct that the sample analyzed in the manuscript is limited. In total, 6.1 and 13.6 hours of low-level observations were performed during AFLUX and ACLOUD, respectively, including measurements over the marginal sea ice zone that were not taken into account here. We agree that additional, statistically representative data sets could help to solidify the conclusions. Therefore, we added the following sentences to the manuscript:

**"Nevertheless, the conclusions are based on limited airborne samples, which might be biased by flight strategy. It would be useful to apply the described method to further, statistically more robust data sets to extensively investigate the impact of changing cloud and environmental properties on the solar CRE."** (line 297–299)

*Specific Comments:*

1. *L95: Can you provide more information on the profiles? How far away are the soundings? Where they found to be comparable to the aircraft data? How did you merge these data?*

The thermodynamic observations during the aircraft ascents and descents form the basis of the thermodynamic profiles, as they were performed locally, confining the respective low-level leg. Only for levels above the maximum flight altitude of the Polar 5 aircraft (usually around 3 km), the thermodynamic state is obtained from the measurements of the radiosonde launched at Ny-Ålesund. The maximum spatial and temporal distance to these radiosoundings is 430 km and 6 hours, respectively. To clarify this strategy, the last sentence of this paragraph is modified to:

**"These profiles resulted from thermodynamic measurements during local aircraft ascents and descents adjacent to the low-level flight sections that were complemented by the radiosonde observations at Ny-Ålesund (max. 430 km away) for atmospheric layers above the maximum flight altitude."** (lines 101–103)

We did not compare the thermodynamic aircraft and radiosonde measurements. On the one hand, we already applied a local thermodynamic profile for the lower atmosphere, making a comparison to the remote radiosonde observations redundant. On the other hand, the lack of local thermodynamic measurements for higher altitudes makes a comparison impossible. Anyway, despite the impact of water vapour on solar radiation, we expect a potential water vapour difference between aircraft and radiosonde location, especially at higher altitudes, to affect the solar CRE at the surface only weakly.

2. *Can you show comparisons between the simulations and observations when the sky was clear to validate the CF estimates? This will help ensure there is not a mean bias in the CRE.*

Figure 1 compares the measured and simulated solar downward irradiance in cloud-free conditions during AFLUX and ACLOUD as a two-dimensional histogram. As the majority of the data points are distributed around the one-to-one line and the coefficient of determination is close to 1, there is a good agreement between the simulation and observation without any mean bias.

We added one sentence to the text: **"For cloud-free conditions, a comparison between measured and simulated $F^{\downarrow}$ yields a coefficient of determination $R^2$ of 0.9971, indicating the accuracy of the simulations."** (lines 103–104)

[Figure]

*Figure 1: Two-dimensional probability density function depending on simulated and observed downward solar irradiance considering all cloud-free observations (τ less than 1.125) of AFLUX and ACLOUD. The dashed line marks the 1:1-line.*

3. *You state the CRE fluxes are w.r.t. the surface (e.g., line 90), but the observations were made at flight level. As you say at L59-60, water vapor is impactful on the solar, and later at L248 that you flew above some cloud layers. You should use the model to assess whether an atmospheric correction is needed to transfer the observations to the surface, and apply it if it is determined to be significant. Otherwise, you should refer to the calculations as what they are, observations from flight level (and calculate accordingly).*

The statement of the importance of the water vapour mentioned in Lines 59-60 holds for the solar downward irradiance in cloud-free conditions excluding the impact of the usually dominating SZA. For the variability of the solar CRE, water vapour only plays a minor role. Nevertheless, it is correct that the water vapour amount above and below the measurement altitude affects the solar downward and upward irradiances. To assess the impact of the flight altitude on the solar CRE, radiative transfer simulations depending on SZA, surface albedo, and cloud optical thickness were performed for both the flight altitude and the surface. For the two analyzed campaigns, the resulting differences are shown as histograms in

Figure 2. In all cases, the underestimation of the CRE at flight altitude compared to the surface is weak and does not exceed 2 W m$^{-2}$. The larger underestimation during ACLOUD results from the lower SZA and the higher flight altitude (up to 250 m) compared to AFLUX (up to 100 m). Due to the weak impact, we did not apply an atmospheric correction.

We added the following sentences to the manuscript: **"During low-level flight sections, the CRE was retrieved at flight altitude from a combination of airborne radiation measurements and radiative transfer simulations. The resulting CRE values are considered representative for the surface since the flight altitude was consistently lower than 250 m and, according to radiative transfer simulations, the corresponding atmospheric impact low-biased the CRE by less than 2 W m$^{-2}$ with respect to the surface. The irradiances $F_{\mathrm{cld}}^{\downarrow}$ and $F_{\mathrm{cld}}^{\uparrow}$ in cloudy conditions were measured ..."** (lines 92–96)

[Figure]

*Figure 2: Probability density function of the difference between the solar CRE simulated at flight altitude and at the surface for (a) AFLUX and (b) ACLOUD.*

Correcting for the cloud layers (sea smoke) referred to in the comment would be a more difficult task. As this sea smoke revealed a complicated structure and we don't have any information about its microphysical properties, we were not able to assess the impact on the surface CRE. However, since the downward solar irradiance is primarily determined by the cloud optical thickness and these structures were optically thin, we assume that the additional impact of the sea smoke on the surface REB would have been weak compared to the thicker clouds above flight level. In contrast, the albedo and, thus, the upward component are biased significantly by the sea smoke. To account for this effect, we tried to estimate, how the resulting relative contributions would have changed if the sea smoke had been absent. For this purpose, we added the following sentence to the manuscript:

**"If the sea smoke had not been present and the open ocean albedo had revealed a typical value of 0.06, the relative contribution of $\alpha$ would have reverted to -2.0 % in favour of the SZA contribution."** (lines 267–269)

4. *L111-112: Cloud optical depth is not a property independent from transmissivity. They are different ways of saying the same thing.*

5. *L112: I don't think it is correct to say multiple reflections actually increase cloud transmissivity, though it would confound your ability to calculate an accurate value using the broadband measurements with e.g., Eqs (4,5).*

This reply concerns the previous two comments. Probably, confusion arose, because the cloud transmissivity is usually associated with Lambert–Beer's law. However, this law only describes the transmissivity of the direct solar irradiance. In our study, we are interested in the combined transmissivity of both direct and diffuse solar radiation, which is defined according to Eq. 6 of the manuscript.

Regarding comment 4: Using Lambert–Beer's law, the cloud transmissivity may serve as a quantity equivalent to the cloud optical thickness $\tau$ if the SZA is constant. Nevertheless, also the direct cloud transmissivity calculated in this way is formally a function of both $\tau$ and SZA. The direct+diffuse cloud transmissivity (Eq. 6 of the manuscript) is far from being equivalent to $\tau$, as it depends on SZA and surface albedo in addition to $\tau$. In this way, the cloud

transmissivity is not independent of $\tau$, but $\tau$, in turn, is independent of the cloud transmissivity. This becomes obvious from the definition of $\tau$ as the vertical integral of the volumetric extinction coefficient $b_{\mathrm{ext}}$ between cloud base and cloud top, where $b_{\mathrm{ext}}$ is obtained by integrating the single-particle extinction properties over the entire particle range:

$$b_{\mathrm{ext}}(\lambda) = \int_0^\infty Q_{\mathrm{ext}}(\lambda, r) \cdot A_{\mathrm{proj}}(r) \cdot n(r)\, \mathrm{d}r.$$

The extinction efficiency $Q_{\mathrm{ext}}(\lambda, r)$, with $\lambda$ being the wavelength, can be approximated by 2 in the solar spectral range. The number concentration $n(r)$ and the projected area $A_{\mathrm{proj}}(r)$ of the cloud droplets solely depend on droplet size $r$. Consequently, $b_{\mathrm{ext}}$, and, thus, $\tau$, is only a function of the microphysical cloud properties.

Regarding comment 5: Since scattered photos constitute the diffuse radiation component, they do not affect the direct component anymore. Thus, multiple scattering cannot affect the direct cloud transmissivity (Lambert–Beer's law). However, diffuse solar radiation transmitted through a cloud can still undergo multiple scattering events and modify the direct+diffuse solar downward irradiance below the cloud. Since the intensity of this multiple scattering largely depends on the brightness of the surface (darker surfaces increase the probability of absorption), the direct+diffuse cloud transmissivity (Eq. 6 of the manuscript) additionally depends on the surface albedo.

Consider a case characterized by a SZA of 60° and a low-level cloud in 400–600 m altitude with a LWP of 30 g m⁻². For these conditions, Figure 3 shows the results of radiative transfer simulations as a function of the surface albedo. The dashed lines correspond to cloud-free conditions, while the solid lines correspond to situations below a cloud. The solar downward irradiance below the cloud strongly increases with increasing surface albedo, while the increase in cloud-free conditions is weak (Figure 3a). As a result, the cloud transmissivity is higher for larger albedo values (Figure 3b). Despite the weak impact of the surface albedo on the solar downward irradiance in cloud-free conditions, Figure 3c clearly shows a decrease of its direct fraction for higher albedo values. The enhanced diffuse component can, thus, only result from multiple reflections between surface and atmosphere, which are favoured in the case of a brighter surface. Although, the radiation is completely diffuse below clouds, these enhanced multiple reflections are responsible for the larger cloud transmissivity.

To stress that we refer to the direct+diffuse cloud transmissivity, we added: **"The cloud transmissivity $\mathcal{T}_{\mathrm{cld}}$ accounts for both the direct and diffuse component of the solar downward irradiance. As $\mathcal{T}_{\mathrm{cld}}$ depends on $\alpha$ and $\mu$ in addition to the independent cloud property $\tau$, it is not an ideally suitable quantity to describe the impact of clouds on the CRE."** (lines 119–121)

To make clear that the surface albedo directly affects the strength of these multiple reflections, we change the phrase **"The intensified multiple reflection over brighter surfaces cause ..."** to **"The intensified surface albedo-induced multiple reflections over brighter surfaces cause ..."** (line 121).

[Figure]

*Figure 3: (a) Solar downward irradiance, (b) cloud transmissivity, and (c) direct fraction of the solar downward irradiance as a function of surface albedo.*

6. *L30: This point is somewhat stylistic. Ramanathan et al. (1989) defined "cloud-radiative forcing" (CRF), as the net effect, as your equation states. While the literature is not very consistent about this (partly because at TOA, CRE=CRF), at the surface it is useful to reserve the term CRE for the difference in the downwelling components. Consider changing to "CRF" terminology.*

We agree that some studies distinguish between CRF (downward and upward components) and CRE (only downward components). However, as mentioned by the reviewer, this is not very consistent in the literature and other studies refer to the CRE including both components. Since Eq. 2 of our manuscript clearly defined what we mean by CRE, we decided to not change the terminology. Instead, we extended the phrase **"cloud radiative effect (CRE)"** to **"cloud radiative effect (CRE; also referred to as cloud radiative forcing, CRF)"** (lines 27–28).

7. *L30: I think it would be helpful to the reader to include subscripts denoting the fluxes and CRE (CRF) are solar, even though infrared is not analyzed here. In that way, no one will misinterpret the figures if they aren't paying close attention to the text.*

Among the authors, we agreed to skip the subscript because of conciseness of the equations. We clearly mentioned in the beginning that only the solar component is analyzed. Furthermore, the title of the work explicitly refers to the **"solar radiative effect"** and the relevant captions of and within all figures still contain the undoubted term **"solar"**.

8. *L32: "quantify the REB in cloud and cloud-free conditions" is not correct. The statement should be "quantify the REB difference between all-sky and cloud-free conditions". Correspondingly, the "cld" subscript in Eq. 1 and 2 is also not correct: it should be "all-sky". For example, CRF = ALL − CLR could alternatively be defined as CRF = [CLD − CLR]\*FCC, where FCC is cloud fraction. I realize it may be a matter of semantics for your application.*

Yes, the statement is correct. The sentence refers to the mentioned (cloudy and cloud-free) net irradiances separately and not to the CRE as a whole. Hopefully, this becomes clearer in the updated version of the sentence: **"The net irradiances $F_{\mathrm{net,cld}}$ and $F_{\mathrm{net,cf}}$ represent the REB in cloudy and cloud-free conditions, respectively, ..."** (line 31).

Furthermore, we decided to stick to the term "cloudy" and the subscript "cld". This term is not to be confused with the term "overcast". By "cloudy", we simply mean that any cloud is present in the field of view of the instrument regardless of the cloud fraction. Since we do not analyze completely cloud-free scenes, this is equivalent to "all-sky" conditions. We added the following sentence: **"Here, cloudy conditions refer to situations where a cloud of any fraction is present."** (lines 32–33)

9. *L50: Not all studies find a cooling effect in summer...Miller et al.*

Yes, that is correct. To consider this exception, the text adapted in response of the reviewer's first general comment accounts for this exception: **"Only during summer, a total cooling effect was observed in most cases, when the magnitude of the solar cooling effect surpassed the TIR CRE due to decreasing SZA and surface albedo."** (lines 49–51).

10. *L111: I'm not certain the best place in the text to do this, but somewhere it would be helpful to state that that transmissivity (and tau) are broadband values.*

We agree and added the term **"broadband"** to the place, where the transmissivity is mentioned the first time (line 111). The same was done for the surface albedo (line 97). However, as discussed earlier, the cloud optical thickness has a weak spectral dependence in the solar range. Therefore, we did not explicitly mention that the cloud optical thickness is a broadband quantity.

11. *L155/Figure 3: can you add the symbols (alpha, tau) to the appropriate axis labels? Perhaps also make it clear that the color-coding from (a) is used also in the other panels.*

To address this comment, we produced an updated version of the figure, which is shown in Figure 4 of these replies. We added the symbols $\mu$, $\tau$, and $a$ to the axis labels and the legend. In this way, we get a better connection between the figure and the caption. To clarify the colour coding, we replaced **"(right y-axes)"** by **"(consistent colour coding used throughout the study)"** in line 160 in the text and extended the now third-last sentence of the figure caption by **"..., applying the same colour coding as in (a)."**. An additional change to the figure concerns different line styles of the yellow, green, and blue lines to be independent of the colour coding.

[Figure]

*Figure 4: Updated version of the manuscript's Fig. 3: We added the symbols μ, τ, and α to the axis labels and the legend in panel (a) and applied a colour coding to panel (d) to indicate the dominant driver of the CRE (green: cloud optical thickness dominant, blue: surface albedo dominant).*

12. *L160: SZA still has an impact on CRE, so maybe replace "impact" with "dependency". I'm not certain what you mean by this sentence because you state that mu will be neglected and then in the very next sentence you set it to a constant. I understand why you made it a constant, but the second statement seems to contradict the first.*

We agree that the SZA still affects the CRE. We actually meant that the almost absent variability of the SZA during the analyzed flight leg did not really affect the variability of the solar CRE during this section. That's why we set the SZA to a constant value, corresponding to the mean SZA during the flight section. So, we actually neglect the variability of the SZA. To express this in the text, we changed **"... the impact of the SZA on the CRE can be neglected ..."** to **"... the CRE change is not significantly driven by the SZA ..."** (line 168).

13. *L188: I don't think this is obvious at all.*

Maybe, the interpretation of the corresponding figure (Fig. 3d in the manuscript) is not accurate enough. Basically, the quantity with the higher relative impact dominates the evolution of the solar CRE at every given point. To better illustrate that, we colour-coded the background of panel (d) according to the dominant driver of the CRE. Please see the updated figure (Figure 4). To account for this change in the figure caption, we extend the second last sentence by **"... and the background colour indicates the dominant CRE driver"**. In the text, **"From Fig. 3d, it is obvious that the CRE change was mostly controlled by τ over open ocean ..."** was replaced by **"The quantity with the largest relative impact indicates the dominant driver of the CRE evolution, which is highlighted by the background colour in Fig. 3d. The frequent green background for latitudes less than 79.7° demonstrates that the CRE change was mostly controlled by τ over open ocean ..."** (lines 196–198).

*Editorial Comments:*

*L21: "on the one hand" L23;*

We do not see any problem with this formulation and kept it.

*L25: I think you mean "is expected to" not "will"*

Yes, thanks for this comment. We corrected it accordingly (line 25).

*L36: "antagonism" is an odd choice of word. "Due to these opposing effects..."?*

Due to a reorganization of the cloud feedback motivation, the concerned sentence was deleted. However, within the updated text, the phrase **"... the opposing effects of solar and TIR CRE ..."** occurs (line 41).

*L111, elsewhere: "suitable"?*

We changed **"suited"** to **"suitable"** (lines 120 and 210).

*L222-223: I don't understand this sentence. I might be a run-on or something.*

We tried to formulate this sentence in easier words. Based on a larger revision of this section, further information is now included around the sentence concerned:

**"Since Eq. 13 is underconstrained with the two unknown variables $\alpha$ and $\tau$, the possible solutions to it are distributed along the black dashed line in Fig. 4 and include both $(\alpha_1, \tau_2)$ and $(\alpha_2, \tau_1)$. However, the fraction of the partial CRE differences with respect to the total CRE difference (i. e., the relative contributions) are not identical for these solutions. To obtain a unique pair of relative contributions, an additional criterion is introduced, which requires $(\alpha_e, \tau_e)$ to lie on the straight connection line between the two states (black solid line in Fig. 4), parameterized as**

$$\begin{pmatrix} \tau \\ \alpha \end{pmatrix} = \begin{pmatrix} \tau_1 \\ \alpha_1 \end{pmatrix} + s \cdot \begin{pmatrix} \tau_2 - \tau_1 \\ \alpha_2 - \alpha_1 \end{pmatrix}. \tag{14}$$

**By inserting Eq. 14 into Eq. 13, this requirement yields a solution for the parameter $s$ that is used to calculate the final evaluation point $(\alpha_e, \tau_e)$ = (0.49, 7.7). For these values, the partial CRE differences eventually quantify the absolute contributions of cloud and surface, which amount to 4.3 W m$^{-2}$ and 127.9 W m$^{-2}$ (red numbers in Fig. 4) and correspond to relative contributions of 3.3 % and 96.7 %, respectively."** (lines 238–247)

*L266: This sentence lacks clarity.*

We added some more information for clarity and replace the sentence by:

**"Since the method using the total differential can lead to significant uncertainties for too large changes of the drivers, an alternative approach to disentangle their contributions was introduced. This decomposition method is similar to the approximate partial radiative perturbation technique (Taylor et al., 2007) and also applicable to partition the CRE difference between two distinct states into the contributions of the drivers."** (lines 287–290)

*L274-275: "an as" to "a"?*

We changed **"an as simple method"** to **"a similarly simple method"** (line 305).

Added literature

- Cesana, G., Del Genio, A. D., Ackerman, A. S., Kelley, M., Elsaesser, G., Fridlind, A. M., Cheng, Y., and Yao, M.-S.: Evaluating models' response of tropical low clouds to SST forcings using CALIPSO observations, Atmos. Chem. Phys., 19, 2813–2832, https://doi.org/10.5194/acp-19-2813-2019, 2019.
- Lutsko, N. J., Popp, M., Nazarian, R. H., & Albright, A. L.: Emergent constraints on regional cloud feedbacks. Geophys. Res. Lett., 48, e2021GL092934. https://doi.org/10.1029/2021GL092934, 2021.

---

## Author Comment (AC2)

**Replies to comments of Review #2**

We would like to thank the reviewers for their positive and constructive feedback and appreciate their time for extensively reading and commenting on the submitted manuscript. Our replies to the referees' comments are structured as follows:

*Referee's comments in italic – line numbers according to initially submitted manuscript*
Authors' responses in roman – line numbers according to adjusted manuscript.
**Citations from the initial and the adjusted manuscript are given in bold.**

Beyond edits related to the reviews, we applied additional revisions to some text passages after carefully going through the manuscript. For these changes, we refer to the track changes file.

*General*

*Airborne radiation measurements over the Arctic marginal sea ice zone and open ocean near Svalbard are analyzed to investigate the dependence of the cloud radiative effect on solar zenith angle, cloud optical thickness and surface albedo. It is found that the latter has by far the largest effect.*

*The manuscript is in all parts very well written, it is clearly organized and the topic is of large scientific interest for climate research. The paper addresses one of the most uncertain factors in climate projections, namely the impact of clouds on the surface energy budget. The analysis helps to better understand the complex interaction processes between clouds and the surface due to their effects on radiation. Most of the text can be well understood but I suggest adding some explanations for non-experts. Altogether, these suggestions and some further hints to the text are all minor points and I recommend the publication of this very well done work after revision.*

*Revisions*

1. *I think, the description around Figure 4 (Caption and corresponding text) can be improved. The meaning of the black dashed line is somehow unclear to me. I guess the red numbers refer to the full change of the Solar CRE between the Open Ocean state and the Sea Ice state rather than to the change from the Ocean state to the intersection point. But this does not become clear. I did not fully understand the meaning of the evaluation point. Following the given numbers (131.6+0.6 = 132.2, 8.3+124.0=132.3, 127.9+4.3=132.2) the way how we come from the open ocean state to the sea ice state plays no role. This should be stressed, if correct.*

To better understand the following discussion, we also refer to the reply on the next general comment. The two terms on the right-hand side of Eq. 13 are the partial CRE differences, which result from only changing one variable between the two states, while the other variable is kept constant. Eventually, these partial CRE differences should correspond to the absolute contributions of cloud optical thickness $\tau$ and surface albedo $\alpha$ to the given CRE difference (132.2 W m$^{-2}$). Since the partial CRE differences are not constant (as indicated by the coloured numbers in Fig. 4 of the manuscript), the procedure described in Sect. 4.1 identifies a pair of values $(\alpha_e, \tau_e)$ that is referred to as evaluation point and assures that the two partial CRE differences exactly add up to the CRE difference. As the reviewer observed correctly, there are

multiple possibilities to compose the CRE difference. I. a., either changing the surface albedo first (partial CRE change of 131.6 W m$^{-2}$) and the cloud optical thickness second (0.6 W m$^{-2}$), or vice versa (8.3 W m$^{-2}$ for the cloud optical thickness change and 124.0 W m$^{-2}$ for the surface albedo change). However, both cases reveal different relative contributions of the partial CRE differences to the total CRE difference. The surface albedo change covers 99.5 % of the total CRE change in the first case, but only 93.7 % in the second case. Further possible solutions are distributed along the black dashed line in Fig. 4 of the manuscript. This multitude of solutions is caused because Eq. 13 is underconstrained. To obtain a single solution, we introduced the second criterion, which requires the evaluation point to lie on the connection line between both states. The intersection of this connection line and the black dashed line is representatively located in the middle between both states and constitutes the evaluation point. At this evaluation point, the partial CRE changes correspond to the red numbers, which, as all partial CRE changes, account for the full change of one driver between the two states, while the other is constant at its value of the evaluation point. The obtained numbers (127.9 W m$^{-2}$ and 4.3 W m$^{-2}$) correspond to the absolute contributions of surface albedo and cloud optical thickness to the CRE change.

The modified text contains a clearer description of Fig. 4 and the associated method. To more easily refer to the median values of $\tau$ and $\alpha$ for the open ocean and sea ice states as well as the corresponding differences, we first assign the symbols $\tau_1$, $\alpha_1$, $\tau_2$, and $\alpha_2$ to the median values: **"... $\tau_1$ = 8.5 and $\alpha_1$ = 0.10 over open ocean ... and $\tau_2$ = 7.0 and $\alpha_2$ = 0.78 over sea ice ..."** (lines 221–222). Subsequently, we revised Sect. 4.1 starting from its second paragraph:

**"The decomposition of $\Delta\mathrm{CRE}$ into partial CRE differences that only account for a change in $\tau$ or $\alpha$ between the two states is given by:**

$$\Delta\mathrm{CRE} = \Delta\mathrm{CRE}_{\Delta\tau}(\alpha) + \Delta\mathrm{CRE}_{\Delta\alpha}(\tau), \tag{13}$$

**which is equivalent to integrating Eq. 10 between the states. For example, the partial CRE difference $\Delta\mathrm{CRE}_{\Delta\tau}(\alpha)$ represents the CRE contrast resulting from a change $\Delta\tau$ from $\tau_1$ to $\tau_2$ at any constant $\alpha$. Due to the non-linear sensitivity of the CRE to both $\tau$ and $\alpha$, $\Delta\mathrm{CRE}_{\Delta\tau}(\alpha)$ and $\Delta\mathrm{CRE}_{\Delta\alpha}(\tau)$ depend on the concrete value that $\alpha$ and $\tau$, respectively, are fixed to. These non-linearities are indicated by the pairs of green and blue numbers in Fig. 4. Despite an identical $\Delta\tau$, the associated $\Delta\mathrm{CRE}_{\Delta\tau}$ is 8.3 W m$^{-2}$ if $\alpha$ corresponds to $\alpha_1$, but only 0.6 W m$^{-2}$ for $\alpha_2$. Similarly, $\Delta\mathrm{CRE}_{\Delta\alpha}$ amounts to 131.6 W m$^{-2}$ if $\tau = \tau_1$ and 124.0 W m$^{-2}$ for $\tau = \tau_2$. Consequently, for neither $(\alpha_1, \tau_1)$ nor $(\alpha_2, \tau_2)$, the partial CRE differences do exactly add up to $\Delta\mathrm{CRE}$ in Eq. 13. Therefore, the approach suggested in the following, which was not considered by the APRP method (Taylor et al. 2007), identifies the values $\alpha_e$ and $\tau_e$ that precisely satisfy Eq. 13. This pair of values is referred to as evaluation point in the following.**

**Since Eq. 13 is underconstrained with the two unknown variables $\alpha$ and $\tau$, the possible solutions to it are distributed along the black dashed line in Fig. 4 and include both $(\alpha_1, \tau_2)$ and $(\alpha_2, \tau_1)$. However, the fraction of the partial CRE differences with respect to the total CRE difference (i. e., the relative contributions) are not identical for these solutions. To obtain a unique pair of relative contributions, an additional criterion is introduced, which requires $(\alpha_e, \tau_e)$ to lie on the straight connection line between the two states (black solid line in Fig. 4), parameterized as**

$$\begin{pmatrix} \tau \\ \alpha \end{pmatrix} = \begin{pmatrix} \tau_1 \\ \alpha_1 \end{pmatrix} + s \cdot \begin{pmatrix} \tau_2 - \tau_1 \\ \alpha_2 - \alpha_1 \end{pmatrix}. \tag{14}$$

**By inserting Eq. 14 into Eq. 13, this requirement yields a solution for the parameter $s$ that is used to calculate the final evaluation point $(\alpha_e, \tau_e)$ = (0.49, 7.7). For these values, the partial CRE differences eventually quantify the absolute contributions of cloud and surface, which amount to 4.3 W m$^{-2}$ and 127.9 W m$^{-2}$ (red numbers in Fig. 4) and correspond to relative contributions of 3.3 % and 96.7 %, respectively."** (lines 226–247)

The caption of Fig. 4 is updated to: **"Solar CRE parameterized with Eq. 9 as a function of $\tau$ and $\alpha$. The symbols indicate the median states $(\alpha_1, \tau_1)$ over open ocean and $(\alpha_2, \tau_2)$ over sea ice calculated for the case study on 4 April 2019. The blue and green numbers (all in W m$^{-2}$) quantify the partial solar CRE change along the respective lines. The red numbers represent the finally obtained absolute contributions of $\tau$ and $\alpha$ to the solar CRE difference, corresponding to the partial solar CRE differences at the evaluation point $(\alpha_e, \tau_e)$. The evaluation point is determined by the intersection of two criteria: first, it must lie on the black solid line connecting the two states and, second, it must satisfy Eq. 13, which is the case for all $(\alpha, \tau)$ along the black dashed line. See text for more details."**

2. *Perhaps, a Discussion section could be added where, e.g. the differences between both methods based on equations (10) and (13) are discussed and their different ranges of validity. In this connection Lines 195-210: Is it possible to give a threshold for the validity of (10)? Further possible points for a discussion is the difference to mid latitudes and if the results have any effects on or benefits for modelling.*

Although we decided not to add a dedicated Discussion section to our manuscript, we tried to cover the suggested discussion points in the updated version of our manuscript.

(1) Differences between methods based on Eqs. 10 and 13:

The first method based on Eq. 10 expresses a change of the CRE by means of the total differential, with

$$\mathrm{dCRE} = \frac{\partial \mathrm{CRE}}{\partial \tau}(\tau, \alpha) \cdot \mathrm{d}\tau + \frac{\partial \mathrm{CRE}}{\partial \alpha}(\tau, \alpha) \cdot \mathrm{d}\alpha,$$

where the contributions of cloud optical thickness $\tau$ and surface albedo $\alpha$ are given by the respective terms on the right-hand side. However, due to the assumption of infinitesimal differences $\mathrm{dCRE}$, $\mathrm{d}\tau$, and $\mathrm{d}\alpha$, the applicability of this method is limited. Actual measured or modelled samples are a discrete collection of data points with ideally small, finite differences $\delta\mathrm{CRE}$, $\delta\tau$, and $\delta\alpha$ between two adjacent points, such that a more accurate formulation of the above equation would be:

$$\delta\mathrm{CRE} = \frac{\partial \mathrm{CRE}}{\partial \tau}(\tau, \alpha) \cdot \delta\tau + \frac{\partial \mathrm{CRE}}{\partial \alpha}(\tau, \alpha) \cdot \delta\alpha + \mathrm{Res}.$$

The finite differences in combination with the non-linear dependence of the CRE on $\alpha$ and $\tau$ (see also Fig. A1 in the manuscript) lead to differences of the sensitivity coefficients $\frac{\partial \mathrm{CRE}}{\partial \tau}$ and $\frac{\partial \mathrm{CRE}}{\partial \alpha}$ between the two neighbouring data points, resulting in the residual term Res. For sufficiently small $\delta\mathrm{CRE}$, $\delta\tau$, and $\delta\alpha$, these sensitivity differences and the residual are minor. The example time series in Fig. 3 of the manuscript has a sampling frequency of 20 seconds and reveals a negligible residual of around 10$^{-6}$ W m$^{-2}$ s$^{-1}$ compared to the CRE change in the order of 0.5 W m$^{-2}$ s$^{-1}$. Therefore, we omit the residual and work with the differential equation in the manuscript. However, coarsening the resolution of the time series to minutely values increases the residual to roughly 0.1 W m$^{-2}$ s$^{-1}$, which

corresponds to 10–20 % of the CRE change and highlights the limits of the method. Note that the resolution of a data set is less crucial than the differences of the sensitivities between adjacent data points. Therefore, the validity of the method using Eq. 10 should be evaluated based on the variability of the sensitivities and the impact of the residual rather than a fixed threshold. Furthermore, just like the example time series, the analysis of data with significant small-scale variability might benefit from a carefully chosen smoothing to reduce the point-to-point differences.

For cases, where the first method is not suitable due to a significant residual, we propose the second method. A given CRE difference ΔCRE between two points (the symbol Δ indicates a too large difference to reliably apply Eq. 10) can be obtained by integrating the total differential:

$$\int_{\mathrm{CRE}_1}^{\mathrm{CRE}_2} \mathrm{dCRE} = \int_{\tau_1}^{\tau_2} \frac{\partial \mathrm{CRE}}{\partial \tau}(\tau, \alpha) \cdot \mathrm{d}\tau + \int_{\alpha_1}^{\alpha_2} \frac{\partial \mathrm{CRE}}{\partial \alpha}(\tau, \alpha) \cdot \mathrm{d}\alpha,$$

where the subscripts 1 and 2 describe the two data points. This integration is equivalent to Eq. 13:

$$\Delta\mathrm{CRE} = \Delta\mathrm{CRE}_{\Delta\tau}(\alpha) + \Delta\mathrm{CRE}_{\Delta\alpha}(\tau) \left[+ \mathrm{Res}\right].$$

However, depending on the exact values used for $\alpha$ and $\tau$, there might still be a residual, which is removed by the procedure described in Sect. 4 of the manuscript (see also discussion on comment above). Since this method is applicable to any difference of $\tau$ and $\alpha$ between two points, it is also suitable to partition a CRE difference between two arbitrary states into its contributions. These states, which may even be averages of multiple data points, can be different points in time and location or the results of two simulations with different input. Although it would be possible to compare two adjacent points of a data series using Eq. 13, the method based on Eq. 10 is sufficient in most cases.

In the manuscript, we considered the above discussion as follows: To better prepare the reader for the discussion, we slightly revised the beginning of Sect. 3.2:

**"For continuous observations with weak differences of the drivers between neighbouring data points, the total differential of Eq. 9 (neglecting the SZA dependence), with**

$$\mathrm{dCRE} = S_\tau(\tau, \alpha) \cdot \mathrm{d}\tau + S_\alpha(\tau, \alpha) \cdot \mathrm{d}\alpha, \tag{10}$$

**yields an accurate result for the corresponding change of the CRE. The terms on the right-hand side of Eq. 10 represent the absolute contributions of $\tau$ and $\alpha$ to the CRE change, which are determined by both the sensitivities of the CRE with respect to $\tau$ ($S_\tau$) and $\alpha$ ($S_\alpha$) and the absolute change of these parameters ($\mathrm{d}\tau$ and $\mathrm{d}\alpha$). The sensitivity coefficients, given by**

$$S_\tau(\tau, \alpha) = \frac{\partial \mathrm{CRE}}{\partial \tau} \text{ and}$$

$$S_\alpha(\tau, \alpha) = \frac{\partial \mathrm{CRE}}{\partial \alpha},$$

**both depend on $\tau$ and $\alpha$ and are discussed in detail in Appendix A. Along the flight leg of the example case, the results of the separated contributions are shown in Fig. 3. The temporal changes of the absolute contributions of $\tau$ and $\alpha$ are illustrated in Fig. 3c and indicate their respective tendency to the CRE transition."** (lines 173–183)

The main discussion on the difference between the methods is added to the beginning of Sect. 4:

**"Due to the assumption of infinitesimal differences in Eq. 10, the approach described in Sect. 3.2 may lead to significant uncertainties if the differences of $\tau$ and $\alpha$ between two data points become too large. This is particularly the case, when the non-constant sensitivity coefficients $S_\tau$ and $S_\alpha$ (see Appendix A) vary significantly between the two points, causing a considerable discrepancy between the CRE change (left-hand side of Eq. 10) and the sum of the absolute contributions (right-hand side of Eq. 10). In this case, another method, which is proposed in the following and applicable to any point-to-point difference, may be considered. This method is likewise suitable for disentangling the contributions of the drivers to a CRE change between two isolated states, such as different points in time or location."** (lines 205–211)

Furthermore, we added the following sentences to the Conclusions section:

**"Since the method using the total differential can lead to significant uncertainties for too large changes of the drivers, an alternative approach to disentangle their contributions was introduced. This decomposition method is similar to the approximate partial radiative perturbation technique (Taylor et al., 2007) and also applicable to partition the CRE difference between two distinct states into the contributions of the drivers."** (lines 287–290)

(2) Difference to mid-latitudes

Since our method is based on a physical relationship between the CRE and the drivers, the method itself is not restricted to the Arctic. However, the relative contributions of the drivers to a CRE change likely differ between the polar regions and the mid-latitudes, e. g., due to less severe surface albedo differences. To account for this discussion, we added a corresponding sentence as an outlook to the Conclusions section:

**"The general approach used in this study is not limited to the Arctic. Since the method is universally applicable to quantify the contributions of drivers to any given CRE difference, it could also be used to assess how the importance of certain drivers differs, e. g., between the polar regions and the mid-latitudes, where surface albedo contrasts are usually weaker."** (lines 299–302)

(3) Effects on models

The method described in Sect. 4 of the manuscript may furthermore help to interpret potential biases of the CRE between simulations and observations. Decomposing the CRE differences quantifies the individual contributions of the drivers to the bias. The results will inform about, which model parameters cause the largest CRE uncertainty and require the most crucial accuracy.

We add the following sentence to the conclusion section: **"Furthermore, modelling could possibly benefit from quantifying the contributions of the drivers to a potential CRE bias, which can help to evaluate for which parameters an accurate representation in the model is most crucial."** (lines 302–304)

*Hints for text improvement*

*Lines 10-12: at this point it is perhaps unclear how non-cloud conditions can dominate the cloud radiative effect.*

We agree that the dominance of the non-cloud properties for CRE differences might be difficult to understand here, although we mentioned before (line 6-7) that the CRE is also affected by solar zenith angle and surface albedo. However, we won't be able to give a full explanation of the relationship between CRE and non-cloud properties already in the abstract and simply stated what we found in the study. Nevertheless, we attempted to clarify the dominating impact of the non-cloud properties by rewriting the last sentence of the abstract as follows:

**"Using the same approach, the analysis is extended to observations from a series of aircraft campaigns and indicates that the variability of the non-cloud properties SZA and surface albedo between seasons and surface types, respectively, has a larger impact on the resulting difference of the solar CRE than the variability of cloud properties."** (lines 8–11)

*Line 22: define a larger/smaller REB*

We see that the meaning of a larger/smaller REB might be unclear at this point as we haven't introduced the corresponding equations yet. To avoid confusion, we skipped "a larger REB" at this point but kept the resulting surface warming as the most important statement. The updated sentence reads:

**"On the one hand, the darker open ocean directly affects the solar REB by increasing the absorption of solar radiation, which leads to an intensified surface warming..."** (lines 21–22)

*Line 41: a negative CRE change is a decrease of the CRE?*

Correct. We added **"...,  i. e., decreasing CRE, ..."** (line 40)

*Line 52: better: solar cooling effect caused by clouds over open ocean?*

We used the formulation **"cooling effect of clouds over open ocean"** (lines 52–53).

*Line 67: unclear: which parameters?*

We agree that the connection to the cloud, surface, and thermodynamic parameters mentioned in the previous sentence, where they are called **"conditions"**, is not obvious. To be consistent, we changed the term **"conditions"** to **"properties"** (line 59) and likewise replace the confusing **"parameters"** by **"considered properties"** (line 61).

*Line 105: why is the index of the transmissivity of cloud free atmosphere 'atm' and not 'cf' ?*

The subscript "cld" does not account for the transmissivity of the entire cloudy atmosphere, but only to the cloud itself. The transmissivity of the atmosphere, but without the cloud, still affects the irradiance in cloudy conditions (see Eq. 9 in the manuscript). This is in contrast to the "cld"

and "cf" net irradiances, which result from interactions of radiation with cloudy (clouds as well as water vapour, aerosols, …) and cloud-free (only water vapour, aerosols, …) atmosphere, respectively. In this way, the subscript "cld" can serve as an addition to "atm" for the transmissivity rather than a counterpart. Therefore, we decided to keep the subscript "atm". Instead, we replace the phrase **"… the transmissivities of cloud $\mathcal{T}_{cld}$ and cloud-free atmosphere $\mathcal{T}_{atm}$ …"**, which might have led to confusion, by the hopefully clearer phrase **"… the broadband transmissivities of cloud $\mathcal{T}_{cld}$ and atmosphere (excluding clouds) $\mathcal{T}_{atm}$ …"** (lines 111–112).

*Line 111: function **of** alpha and mu*

Correct. Thanks for noticing. However, we changed the formulation of the sentence such that this word group does not occur anymore.

*Line 143: better transition of the CRE from … to …*

We added **"…, e. g., from open ocean to sea ice, …"** (line 151–152)

*Caption figure 3: the caption text should better fit to what is used in the figure, so either you use abbreviation mu, tau alpha or the complete names. Also, please add the day to which these observations belong.*

Based on a comment of the other reviewer, we added the symbols $\mu$, $\tau$, and $\alpha$ to the axis labels and the legend in Fig. 3a, which improves the connection between the figure and the caption. Furthermore, we changed the line style of the yellow, green, and blue lines to be independent of the colour coding and coloured the background of panel (d) to indicate the dominant driver of the CRE transition. The updated version of this figure is shown in Figure 1 of these replies. Additionally, we mentioned the day by adding **"… flight leg performed on 4 April 2019"** to the end of the first sentence of the figure caption.

*Line 195: better: two states defined by different values of optical thickness, surface albedo etc.*

Due to a revision of the introduction to Sect. 4, the sentence concerned no longer occurs in the updated manuscript (please also see the discussion on the first point of the second general comment).

*Line 195: without correlation means here simply large differences between the states?*

Yes, basically that is what we wanted to express. For too large differences of $\tau$ and $\alpha$ between adjacent data points, the method based on Eq. 10 is not suitable. Please see the discussion on the first point of the second general comment for more details and the implemented text revisions.

[Figure]

*Figure 1: Updated version of the manuscript's Fig. 3: We added the symbols μ, τ, and α to the axis labels and the legend in panel (a) and applied a colour coding to panel (d) to indicate the dominant driver of the CRE (green: cloud optical thickness dominant, blue: surface albedo dominant).*

*Please increase the thickness of the green line in the figure.*

We agree that especially the green lines are badly visible. To be consistent, we thickened all lines. Additionally, we framed the coloured numbers for better contrast (especially for the number 4.3). The updated Fig. 4 of the manuscript is shown in Figure 2 of this document.

*Line 215: perhaps better: see the pairs of green blue and red numbers ....*

After rewriting the corresponding section, the phrase **"pairs of green and blue numbers in Fig. 4"** now occurs in line 232.

[Figure]

*Figure 2: Updated version of the manuscript's Figure 4 with thicker green, blue, and red lines*